

# Base synergy in freshly nucleated particles

Galib Hasan, Haide Wu, Yosef Knattrup, and Jonas Elm

Department of Chemistry, Aarhus University, Langelandsgade 140, 8000 Aarhus C, Denmark

**Correspondence:** Jonas Elm (jelm@chem.au.dk)

**Abstract.** Sulfuric acid (SA), ammonia (AM) and dimethylamine (DMA) are believed to be key contributors to new particle formation (NPF) in the atmosphere. NPF happens through gas-to-particle transformation via cluster formation. However, it is not obvious how small clusters grow to larger sizes and eventually form stable aerosol particles. Recent experimental measurements showed that the presence of mixtures of bases enhance the nucleation rate several orders of magnitude. Using quantum

chemistry methods, this study explores this base synergy in the formation of large clusters from a mixture of SA, AM, and DMA. We calculated the binding free energies of the $(SA)_n(AM)_x(DMA)_{n-x}$ clusters, with $n$ from 1 to 10, where $x$ runs from 0 to $n$. The cluster structures were obtained using our recently developed comprehensive configurational sampling approach based on multiple ABCluster runs and metadynamics sampling via CREST. The structures and thermochemical parameters are calculated at the B97-3c level of theory. The final single point energy of the clusters is calculated at the $\omega$B97X-DJB3/6-

311++G(3df,3pd) level of theory.

Based on the calculated thermochemistry, we found that AM, despite being a weaker base, forms more intermolecular interactions than DMA and easily becomes embedded in the cluster core. This leads to the mixed SA-AM/DMA clusters being lower in free energy compared to the pure SA–AM and SA–DMA clusters. We find that the strong base DMA is important in the very initial steps in cluster formation, but for larger clusters an increased ammonia content is found. We also observed that

the cluster-to-particle transition point for the mixed SA–AM–DMA clusters occurs at a cluster size of 14 monomers, which is notably smaller than the transition points for the pure SA-AM (16 monomers) or pure SA–DMA (20 monomers) systems. This indicates a strong synergistic effect when both AM and DMA are present, leading to the formation of stable freshly nucleated particles (FNPs) at smaller cluster sizes. These findings emphasize the importance of considering several base molecules, when studying the formation and growth of FNPs.

## 20  1  Introduction

Atmospheric aerosols, particularly fine particles (<1 $\mu$m in diameter) and ultrafine particles (<100 nm in diameter), significantly impact human health by being the primary contributors to air pollution-related mortality (Pelucchi et al., 2009; Cromar K, 2023). Aerosols also directly influence the Earth's energy budget by scattering and absorbing solar radiation, resulting in cooling and warming effects, respectively (Loeb and Kato, 2002). Additionally, they have an even larger indirect effect on

the global climate by serving as cloud condensation nuclei (CCN) for the formation of clouds, fog, or mist (Rosenfeld et al., 2014). To date, the interactions between aerosol particles and clouds remain the least understood process in global climate





estimation (Cooley et al., 2023). Primary aerosols are directly emitted into the atmosphere, while secondary aerosols are formed through gas-to-particle conversion, resulting in the formation of freshly nucleated particles (FNPs). The formation of aerosols is initiated by forming strong hydrogen bonded molecular clusters from different atmospheric vapor molecules

(Kulmala et al., 2013). Clusters that possess strong intermolecular interactions can be stable against evaporation, and further grow into aerosol particles of roughly 2 nm and above. Inorganic acids, such as sulfuric acid, and bases, such as ammonia and amines, are key components in the initial cluster formation in the atmosphere (Spracklen et al., 2006; Sipilä et al., 2010; Kirkby et al., 2011; Almeida et al., 2013). In addition, other chemical constituents are also believed to influence clustering such as ions originating from galactic cosmic rays (Kirkby et al., 2016) and condensation of highly oxygenated molecules

(HOMs) (Bianchi et al., 2016). To understand the initial stages of atmospheric aerosol formation, it is essential to know the concentrations and chemical composition of the clusters along with the gaseous compounds that contribute to their growth. Measurements of clusters below 2 nm are extremely challenging, and no comprehensive, simultaneous field measurements of these clusters and their precursors have been conducted until now (Kulmala et al., 2013). Standard condensation particle counters (CPC) typically have detection thresholds around 2-3 nm, making them inadequate for detecting the smallest clusters

(McMurry, 2000). While particle size magnifiers (PSM) can detect clusters as small as ∼1.5 nm, they are expensive, have poor counting efficiency, and do not provide information about the chemical composition of the clusters (Vanhanen et al., 2011). Techniques such as the chemical ionization atmospheric pressure interface mass spectrometer (CI-APi-TOF) are necessary to determine the chemical composition of growing clusters (Jokinen et al., 2012). However, these techniques alter the clusters' composition due to fragmentation during the measurement process, leading to potentially inaccurate results (Zapadinsky et al.,

2018; Passananti et al., 2019; Alfaouri et al., 2022). The CI-APi-TOF can usually only detect clusters up to a certain size of around 10 acid molecules and 10 base molecules Almeida et al. (2013). This leaves a knowledge gap in the chemical composition of large clusters in the range of 1.0 nm to 2.0 nm. As this is the size range for cluster stabilization, it is crucial to get a better understanding of this unknown cluster-to-particle transition regime (Kulmala et al., 2013). Wu et al. (2023) recently presented a robust computational framework that can be applied to study clustering from single molecules all the way

up to 2 nm clusters. Such an approach offers an improved understanding of the chemical interactions and stability of these atmospheric clusters.

Ammonia (AM) and Dimethylamine (DMA) are key contributors to the initiation of sulfuric acid (SA) nucleation and greatly enhance the particle formation rates compared to the pure sulfuric acid or sulfuric acid–water systems (Sipilä et al., 2010; Almeida et al., 2013). This happens due to the proton transfer reactions between the acid and base molecules which

leads to salt formation. Numerous quantum chemical studies have corroborated the role of bases such as AM (Ianni and Bandy, 1999; Larson et al., 1999; Nadykto and Yu, 2007; Kurtén et al., 2007; Loukonen et al., 2010; Herb et al., 2011; DePalma et al., 2012, 2014) and DMA (Kurtén et al., 2008; Loukonen et al., 2010; Kupiainen-Määttä et al., 2012; Ortega et al., 2012; Olenius et al., 2013b; Nadykto et al., 2014; Henschel et al., 2014; DePalma et al., 2012, 2014; Henschel et al., 2016; Ma et al., 2016) in the stabilizing the initial SA clusters.

Recent experimental studies have demonstrated that the simultaneous inclusion of both ammonia and amines with sulfuric acid increases new particle formation rates by 10–100 times compared to mixtures containing only sulfuric acid and amines



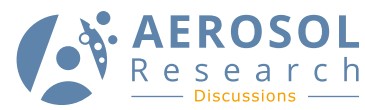

(Glasoe et al., 2015; Yu et al., 2012). This effect cannot be explained by the aqueous-phase base constant or the gas-phase proton affinity, suggesting that the underlying reason for this base synergy is not well understood. One possible explanation is that atmospheric ammonia concentrations are usually much higher than those of amines. However, laboratory experiments

demonstrated that when ammonia levels were lower than dimethylamine, the reactions between sulfuric acid and bases still produced nanoparticles with a higher ammonia content compared to dimethylamine (Lawler et al., 2016). This observation suggests that ammonia uptake is driven by a physicochemical effect in the early stages of particle formation, rather than by the relative concentrations of the substances.

Temelso et al. (2018) provided theoretical evidence of base synergy by calculating free energies of three-component molec-
ular clusters composed of sulfuric acid, amine (dimethylamine or trimethylamine), and ammonia. Their findings indicated that adding sulfuric acid to a cluster containing these mixed bases is thermodynamically more favorable than adding it to a cluster with only sulfuric acid and a single type of amine. Myllys et al. (2019) extended the work to larger clusters and provided a molecular-level explanation for the synergistic effects in SA–AM–DMA cluster formation, showing that ammonia can act as a "bridge-former" and is more likely to be protonated than dimethylamine despite having lower gas-phase basicity. Their
quantum chemical simulations indicated that ammonia's inclusion can increase the particle formation rate by up to 5 orders of magnitude compared to the SA–DMA system. However, these studies have been limited to very small clusters of up to eight monomers and thereby does not give insight into the cluster-to-paritcle transition point of the mixed SA–AM–DMA clusters, nor the AM to DMA ratio in the growing clusters.

Previously, our group has pushed the boundaries of studying large $(SA)_n(AM)_n$ clusters, investigating systems with up
to 60 molecules ($n = 30$) to understand the transition from clusters to particles (Engsvang and Elm, 2022; Engsvang et al., 2023). The exponential increase in the number of possible configurations with respect to cluster size required an improved configurational sampling approach as described by Wu et al. (2023). In this work we extended our previous efforts on studying large clusters and perform quantum chemical (QC) calculations on mixed $(SA)_n(AM)_x(DMA)_{n-x}$ clusters, with $n$ from 1 to 10 and $0 \leq x \leq n$. Hence, we study clusters with an acid-to-base ratio of 1:1 and all combinations of AM and DMA for each
cluster size. We recently proposed a property-based criteria for defining "freshly nucleated particles (FNPs)", as the boundary between discrete cluster configurations and bulk particles (Wu et al., 2024). Here we extend this concept to clusters with mixed bases. Our study suggests that mixed clusters possess varying stability, with the introduction of AM becoming increasingly favorable as the cluster size grows, facilitating the cluster-to-particle transition process leading to FNPs at an earlier stage.

## 2   Methods

### 2.1   Computational details


Density functional theory calculations during the configurational sampling procedures (single point energies, geometry optimization and vibrational frequency calculations) were performed using the empirically corrected B97-3c method (Brandenburg et al., 2018) in the ORCA 5.0.4 quantum chemistry program (Neese, 2022). Single point energies using $\omega$B97X-D3BJ (Najibi and Goerigk, 2018) with 6-311++G(3df,3dp) (Ditchfield et al., 1971), ma-def2-QZVPP (Zheng et al., 2011), ma-def2-TZVP





(Zheng et al., 2011) and def2-TZVDP (Weigend and Ahlrichs, 2005) were also performed in ORCA 5.0.4. The GFN1-xTB (Grimme et al., 2017) and the reparameterized GFN1-xTB$^{\text{re-par}}$ based on previous FNP structures (Wu et al., 2024) semi-empirical calculations were performed using the xTB 6.4.0 program (Bannwarth et al., 2021). The reparameterization was performed according to the workflow given by Knattrup et al. (2024) where the energy and gradients of GFN1-xTB were optimized to fit the FNP structures and energies at the B97-3c level of theory. We switched to the GFN1-xTB$^{\text{re-par}}$ method

once it became available, hence the overall sampling has been performed with a mix of GFN1-xTB and GFN1-xTB$^{\text{re-par}}$. The meta-dynamics calculations were conducted using CREST in non-covalent interaction mode (Pracht et al., 2017, 2020; Pracht and Grimme, 2021; Grimme, 2019; Spicher et al., 2022). Initial clusters were generated with ABCluster version 3.2 (Zhang and Dolg, 2015, 2016) with a CHARMM force field (Huang and MacKerell Jr, 2013).

## 2.2 Configurational sampling workflow

We study the $(\text{SA})_n(\text{AM})_x(\text{DMA})_{n-x}$ clusters, with $n$ from 1 to 10 and $0 \leq x \leq n$. This leads to $n+1$ compositions for each cluster size $n$, implying that we have to sample many of the largest $(\text{SA})_{7-10}(\text{base})_{7-10}$ cluster structures. We employed our recently established configurational sampling protocol, presented by Wu et al. (2023, 2024), demonstrating an excellent balance between accuracy and computational cost. The configurational sampling procedure can be outlined as follows:

$$\text{ABC} \xrightarrow{N=10,000} \text{xTB}^{\text{OPT}} \xrightarrow{N=10,000} \text{B97-3c}^{\text{SP}} \xrightarrow[\text{filter}]{N=1,000} \text{B97-3c}^{\text{PART OPT}} \xrightarrow[\text{filter}]{N=100} \text{B97-3c}^{\text{FULL OPT}} \qquad \text{(ABC track)}$$

Initial cluster structures were generated through 10 parallel ABCluster runs, yielding a total of 10,000 local minima configurations. Our previous studies (Wu et al., 2023, 2024) confirmed that conducting multiple parallel ABCluster explorations provides more accurate predictions for the global energy minimum structures of large clusters compared to a single, prolonged exploration. We used ionic monomers but maintained the overall cluster charge neutral, to facilitate proton transfer, as this is typically observed in the lowest free energy clusters. Subsequent geometry optimizations were performed on all these clusters

using the GFN1-xTB (Grimme et al., 2017) semi-empirical method. DFT single-point calculations were then conducted using the B97-3c method (Brandenburg et al., 2018) on top of the GFN1-xTB optimized conformers.

Next, we filtered out high energy configurations by selecting only the 1,000 lowest structures for partial optimization at the B97-3c level. This partial optimization step was chosen to save computational time and eliminate energetically high-lying configurations. From these, we chose the 100 lowest energy configurations for full optimization and vibrational frequency

calculations.

The lowest free energy conformer was then selected for CREST exploration as suggest by Knattrup et al. (2024), which involves metadynamics to provide reasonably good geometries using the following workflow:

$$\text{CREST} \xrightarrow{N=100} \text{B97-3c}^{\text{FULL OPT}} \qquad \text{(CREST track)}$$

The CREST exploration was performed using GFN1-xTB/GFN1-xTB$^{\text{re-par}}$ in non-covalent interaction mode. We selected the

100 best geometries from the CREST optimization and performed full optimizations and quasi-harmonic vibrational frequency calculations (Grimme, 2012) to obtain the corresponding Gibbs free energies. The overall workflow is highly computationally demanding, but should yield a good estimate of the lowest free energy structures.





## 2.3 Refined Single Point Energies

Engsvang and Elm (2022) benchmarked the ammonia addition and sulfuric acid evaporation energies from the large $(SA)_{10}(AM)_{10}$

FNP. They found that $\omega$B97X-D3BJ/6-311++G(3df,3pd) single-point energies correlated well and gave low error compared to the DLPNO-CCSD(T$_0$)/aug-cc-pVTZ (Riplinger and Neese, 2013; Riplinger et al., 2013) reference values. We have extended this benchmark to investigate the larger basis sets (def2-TZVPD, ma-def2-TZVP and ma-def2-QZVPP) with the $\omega$B97X-D3BJ functional using a subset of the DLPNO reference values calculated by Engsvang and Elm (2022). This dataset contains 216 $(SA)_9(AM)_{10}$ cluster configurations, 279 $(SA)_{10}(AM)_{10}$ and 87 conformers of $(SA)_{10}(AM)_{11}$. The error in electronic binding

energies is shown in Figure 1.

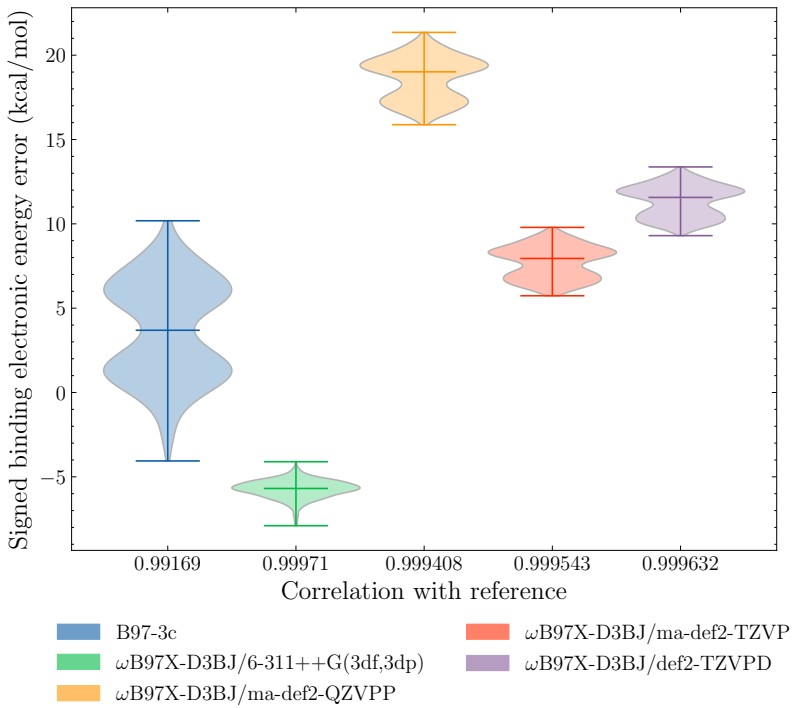

**Figure 1.** Electronic binding energy errors of the $(SA)_9(AM)_{10}$, $(SA)_{10}(AM)_{10}$, and $(SA)_{10}(AM)_{11}$ clusters compared to the DLPNO-CCSD(T$_0$)/aug-cc-pVTZ reference values. The x-ticks show the person correlation value.

In general, there is an improvement, in the form of a lower spread, when applying a higher level single-point energy compared to only using B97-3c. The Def2 basis sets generally lead to too positive binding energies and show a double distribution. The double distribution is also present for B97-3c which is consistent with using a basis set based on the Def2-type basis. The upper

distribution are the $(SA)_{10}(AM)_{10}$ clusters and the lower distribution originates from the $(SA)_9(AM)_{10}$ and $(SA)_{10}(AM)_{11}$ clusters. This is likely due to the clusters with an equal number of base-acid pairs being more favorable yielding a numerical larger free energy and thus a larger relative error.





We furthermore tried the augmented Jensen basis set (pc) (Jensen, 2001, 2002) and augmented correlation consistent (cc) basis set (Dunning, 1989) but the SCF procedure failed to converge for most of the configurations due to linear dependencies in the basis. The few that did converge showed similar large errors to the Def2 basis sets. The emergence of linear dependencies is linked to the addition of many augmented diffuse functions and not the pc or cc basis set themselves.

The large 6-311++G(3df,3pd) Pople basis set shows the lowest binding energy error with a narrow distribution around -5.7 kcal/mol. This is probably due to a fortuitous cancellation of error. We speculate that the error does not show a double distribution because the relative errors are not large enough. Figure 2 presents the errors in addition energies. The addition energies are calculated by the addition of an SA molecule to the $(SA)_9(AM)_{10}$ cluster, yielding $(SA)_{10}(AM)_{10}$, and the addition of an AM molecule to the $(SA)_{10}(AM)_{10}$ cluster, yielding $(SA)_{10}(AM)_{11}$. Reference values are calculated at the DLPNO-CCSD($T_0$)/aug-cc-pVTZ level of theory. "Lowest" refers to the error using the lowest energy conformers at the given level of theory and "DLPNO conformers" uses the conformers lowest at the DLPNO level of theory.

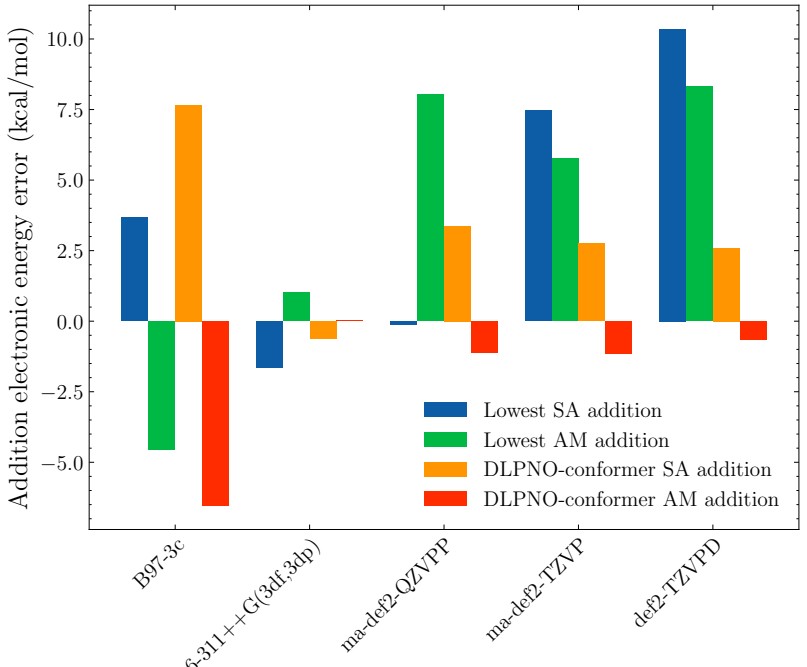

**Figure 2.** Addition electronic energy errors of the addition of SA from $(SA)_9(AM)_{10}$ to $(SA)_{10}(AM)_{10}$, and AM addition from $(SA)_{10}(AM)_{10}$ to $(SA)_{10}(AM)_{11}$ DLPNO-CCSD($T_0$)/aug-cc-pVTZ reference values. "Lowest" is the error using the lowest energy conformers at the given level of theory and "DLPNO conformers" is a comparison between the conformers lowest at the DLPNO level of theory.

Similar to above we see quite large errors for B97-3c, and the $\omega$B97X-D3BJ functional using the def2-TZVPD, ma-def2-TZVP and ma-def2-QZVPP basis sets. Again, we observe the $\omega$B97X-D3BJ functional with the 6-311++G(3df,3pd) to have the lowest errors in all cases. While this is likely a fortuitous cancellation of errors, it appears to be systematic for the systems studied. Based on the previous study by Engsvang and Elm (2022) and the findings here, we will employ the $\omega$B97X-D3BJ/6-





311++G(3df,3pd) for refining the single point energy of the clusters in the current work. However, we emphasize that it would
be worth to test other approaches for obtaining higher level single point energy corrections in the future.

## 2.4 Cluster Binding Free Energies

We calculated the cluster standard binding free energies by subtracting the free energy of the cluster to the sum of the free
energies of the individual monomers. It is calculated as follows:

$$\Delta G_{\text{bind}} = G_{\text{cluster}} - \sum_i G_{\text{monomer},i} \tag{1}$$

In a similar manner the electronic binding energies and the binding thermal correction to the free energy can be calculated.
This allows the division of the binding free energy in the following terms:

$$\Delta G_{\text{bind}} = \Delta E_{\text{bind}} + \Delta G_{\text{bind, thermal}} \tag{2}$$

Here we calculated the structures and thermochemistry at the B97-3c level (the $\Delta G_{\text{bind,thermal}}$ term) and the binding electronic
energy at the $\omega$B97X-DJB3/6-311++G(3df,3pd) level (the $\Delta E_{\text{bind}}$ term). It should be noted that the $\Delta G_{\text{bind, thermal}}$ term is
calculated using the quasi-harmonic approximation (Grimme, 2012) as default in ORCA. The quasi-harmonic approximation
removes spurious low vibrational modes, but does not take local and global anharmonicity into account. The recent work by
Halonen (2024) provides a promising avenue to further improve our understanding of cluster stability, by deriving an analytical
expression to account for local anharmonicity. Integrating this approach in future studies would increase the accuracy of our
thermodynamic predictions, but as it is not straight forward how to obtain the energy barriers between cluster configuration, it
is beyond the scope of the current work.

The equations above account solely for the thermochemistry of the clusters. To calculate the "actual" binding free energies
under specific conditions, we use the self-consistent distribution function (Wilemski and Wyslouzil, 1995; Halonen, 2022):

$$\Delta G_{\text{bind}}(\boldsymbol{p}) = \Delta G_{\text{bind}} - RT \cdot \left(1 - \frac{1}{n}\right) \cdot \sum_i \ln\left(\frac{p_i}{p_{\text{ref}}}\right) \tag{3}$$

Here $p_{\text{ref}}$ corresponds to a reference pressure (1 atm) and $p_i$ represents monomer partial pressures. The self-consistent for-
mulation allow us to correctly establish the monomer free energies as zero. We previously Wu et al. (2024) tested various
formulations of the actual free energies at given conditions and found no deviations between the calculated free energies.

## 3 Results and discussion

### 3.1 Binding Free Energy at Standard Condition

Applying the outlined extensive cluster sampling approach we studied the mixed $(SA)_n(AM)_x(DMA)_{n-x}$ clusters, with $n$
from 1 to 10 and $0 \leq x \leq n$. The pure SA–AM and SA–DMA clusters are taken from (Wu et al., 2024), with refined $\omega$B97X-
D3BJ/6-311++G(3df,3pd) single point energies calculated in this work. Figure 2a presents the standard binding free energies,





calculated at 298.15 K and 1 atm, as a function of the number of monomers $m$ in the cluster. Each point on the graph is labeled with a pair (AM, DMA), indicating the numbers of ammonia and dimethylamine monomers in the cluster.

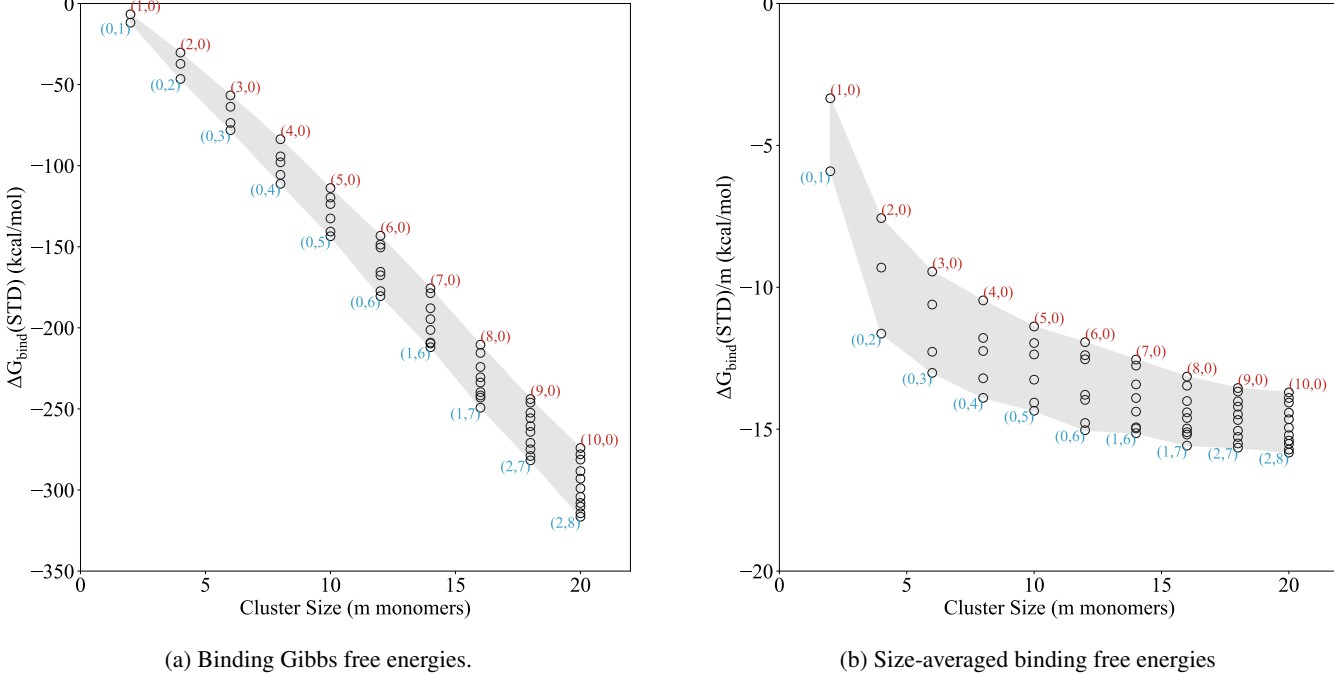

(a) Binding Gibbs free energies.

(b) Size-averaged binding free energies

**Figure 3.** (a) Binding Gibbs free energy and (b) size-averaged binding Gibbs free energy of the $(SA)_n(AM)_x(DMA)_{n-x}$ clusters, with $n$ from 1 to 10 and $0 \le x \le n$., under standard condition (298,15 k and 1 tm). The free energies are calculated at the $\omega$B97X-D3BJ/6-311++G(3df,3pd)//B97-3c level of theory.

Based on the standard binding free energies (Figure 3a), we see that the SA–DMA clusters are the most stable up to 12 monomers. Beyond this point, it is more favorable to exchange 1-2 DMA molecules with AM. Comparing the AM to DMA ratio for smaller clusters, the (5,0) cluster is significantly higher in free energy, by almost 20 kcal/mol, compared to the (0,5) cluster, indicating that DMA alone provides substantial stability for the small cluster sizes.

For cluster sizes around 14–16 monomers, compositions such as (0,7) and (0,8) are correspondingly 2.9 kcal/mol and 9.4 kcal/mol higher in free energies than (1,6) and (1,7), suggesting that introducing one AM molecule increases the stability compared to only having DMA in the clusters. This trend is also observed for larger clusters, where the clusters with compositions (0,9) and (0,10) are higher in free energies compared to (2,7) and (2,8), indicating that having one or two AM molecules in addition to DMA provides higher stability for the larger clusters.

Figure 3b presents the size averaged binding free energies ($\Delta G_{bind}/m$). These values represent the average binding free energy of each molecule in the cluster. Similar to our previous work (Engsvang and Elm, 2022; Engsvang et al., 2023; Wu et al., 2023, 2024) we see that the size-averaged free energy rapidly decrease as a function of cluster size and levels out around 12–20 monomers. This can be interpreted as the cluster transitioning towards more particle-like properties.



Using the convex hull method from Wu et al. (2024), we studied the formation of solvation shells for clusters with the lowest free energy at each size. When the cluster is composed only of SA and DMA (6,0), no encapsulation of ions occurs. However, when AM is added, the lowest free energy structure favors a shell structure, with AM encapsulated by SA and DMA in clusters of size (1,6) to (1,7) (see Figure 4a). When two AM molecules are introduced, a single solvation shell forms, encapsulating both AM monomers as shown in Figure 4b.

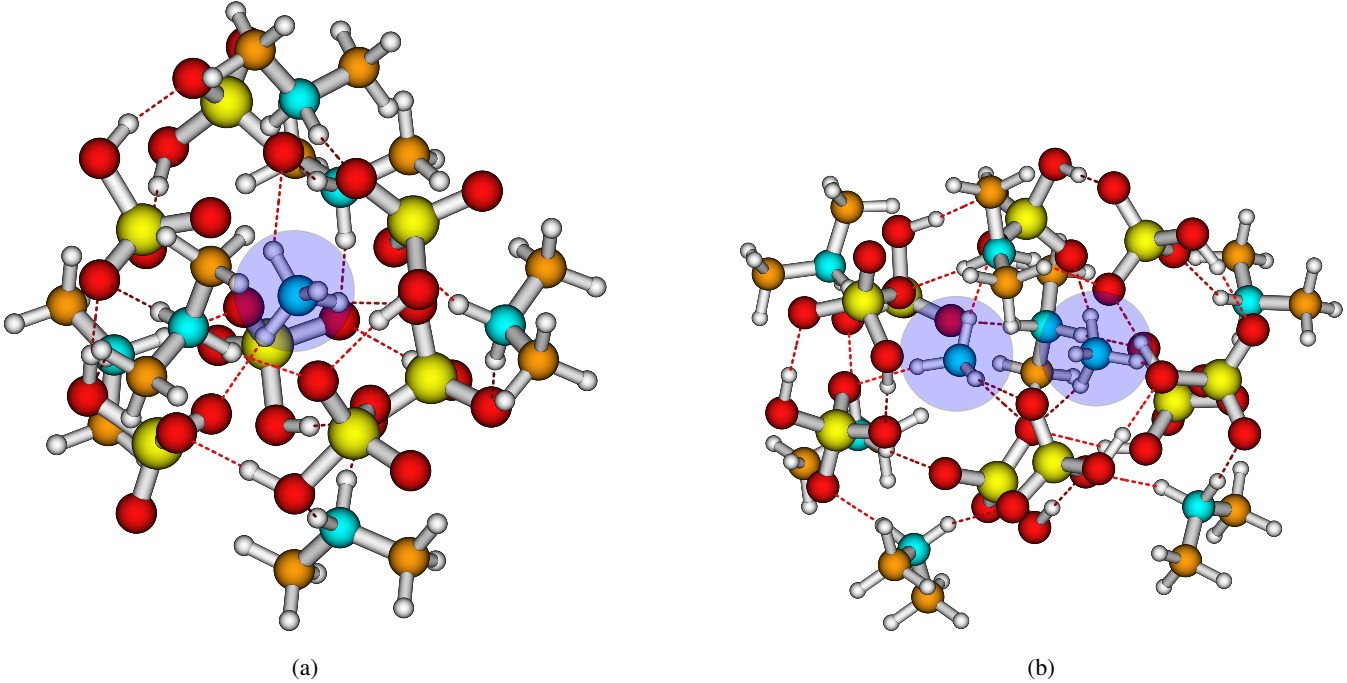

(a)

(b)

**Figure 4.** Two examples of protonated ammonia (blue circle) being encapsulated for (a) $(SA)_7(AM)_1(DMA)_6$, (b) $(SA)_9(AM)_2(DMA)_7$. The structures are the lowest in free energy at the $\omega$B97X-D3BJ/6-311++G(3df,3pd)//B97-3c level of theory. White = hydrogen, blue = nitrogren, yellow = nitrogen, red = oxygen, and brown = carbon.

The reason for the more stable clusters containing 1-2 AM molecules can be ascribed to an intricate combination of hydrogen bond capacity, base strength and steric hindrance. For instance, the $T_d$ symmetry of the AM molecule, makes it capable of forming four intermolecular bonds, whereas DMA can only form two intermolecular bonds. This will become increasingly important as the cluster becomes larger and thus more spherical, as it increases the possible coordination. In addition, the presence of the two bulky methyl groups in DMA impose a strong steric hindrance. As a result, the protonated AM stays embedded at the center of the cluster and forms a fully coordinated complex with the surrounding $HSO_4^-$ ions. This trend suggests that AM starts to play a more significant role in stabilizing larger clusters, and its presence becomes increasingly favorable as the cluster size grows. This is consistent with the experimental work by Lawler et al. (2016) that found increased AM content compared to DMA in newly formed SA–AM–DMA nanoparticles. A consequence of these findings is that the cluster-to-particle transition point differs significantly in the mixed base clusters compared to the pure SA–AM and SA–DMA



clusters. Hence, the cluster-to-particle transition point occurs at a cluster size of 14 monomers for SA–AM–DMA, compared to
220   16 for the SA–AM system and 20 for the SA–DMA system. We note that the cluster-to-particle transition point identified from
our previous work (Wu et al., 2024) was unchanged by the single point refinement single point refinement. Overall, this implies
that there is a synergistic effect between the bases AM and DMA for the formation of freshly nucleated particles (FNPs).

## 3.2   Binding Free Energies at Given Conditions

Based on the calculated binding Gibbs free energies at standard conditions described in previous section, we can evaluate the
225   binding free energies under certain monomer concentration and temperature conditions using equation (3). Figure 5 presents
the binding free energies of the clusters at 278.15 K and 298.15 K. We considered two specific conditions: a low concen-
tration regime ([SA] = $10^6$ molecules cm$^{-3}$, [DMA] = 1 ppt, [AM] = 10 ppt) and a high concentration regime ([SA] =
$10^6$ molecules cm$^{-3}$, [DMA] = 10 ppt, [AM] = 10 ppb). These concentrations and temperatures align with typical conditions
employed in the CLOUD chamber experiments (Almeida et al., 2013; Kürten et al., 2018) and real-world nucleation observa-
230   tions (Kürten et al., 2014). However, we do note that for clean environments the "low conc." regime applied here, might still
represent an upper bound. In the following, we will go through each scenario.



(a) 298.15 K - Low conc.

(b) 298.15 K - High conc.

(c) 278.15 K - Low conc.

(d) 278.15 K - High conc.

**Figure 5.** Binding Gibbs free energy, at the $\omega$B97X-D3BJ/6-311++G(3df,3pd)//B97-3c level of theory of the $(SA)_n(AM)_x(DMA)_{n-x}$ clusters, with $n = x$ between 1 and 10, at given condition of temperature (298.15 K and 278.15 K) and different monomer concentration. Monomer concentrations of [SA] was fixed at $10^6$ molecules cm$^{-3}$. "High conc." refers to [AM] = 10 ppb, [MA] = [DMA] = 10 ppt. "Low conc." refers to [AM] = 10 ppt, [MA] = [DMA] = 1 ppt.





**Low concentrations, 298,15 K**

Similar to the standard free energies in previous section, the actual free energies in Figure 5a, shows that the clusters have lower free energy when they are composed of 1–2 AM molecules for the larger cluster sizes. This scenario is seen from the smallest composition with 14 monomers to the largest cluster composition with 20 monomers. However, the clusters do predominantly contain DMA compared to AM. Across all the studied cluster sizes the clusters with a high AM content are generally higher in binding free energies compared to those with a mix of AM and DMA or higher DMA content. We see that the pure SA–AM clusters have a nucleation barrier with a critial cluster at 8 monomers. This is not the case when the clusters are composed of only SA–DMA and is less pronounced for the SA–AM–DMA clusters with mixed bases. These finding aligns with the experimental results of Glasoe et al. (2015), which demonstrated that the formation of 1.8 nm sulfuric acid–base particles followed the trend: AM < MA < DMA < TMA.

**High concentrations, 298,15 K**

In Figure 5b, we see a much narrower span in the binding free energies at given conditions compared to Figure 5a. This is primarily caused by an increased stability of the SA–AM clusters due to higher concentration of AM. On the contrary the SA–DMA clusters are much less affected by the increased concentration as they already have relatively low evaporation rates. The binding free energies are generally more negative compared to the low concentration regime at 298.15 K, logically indicating greater stability at higher concentrations. This also leads to a lower nucleation barrier in the system, with a smaller critical cluster around 4 monomers. These results are consistent with the previous work by Olenius et al. (2013a), Besel et al. (2020) and Kubecka et al. (2023). Due to the high concentration of AM = 10 ppb, we see AM molecules emerging in the lowest free energy clusters at much smaller sizes such as in (1,4) and (1,5). In a similar manner two AM molecules are also found in the cluster at smaller sizes of (2,5). We also see the emergence of three AM molecules in the $m = 16$ cluster. Interestingly, the most stable combination of AM and DMA changes from (3,5) to (2,7) when growing from 16 to 18 monomers. This implies the evaporation of 1 AM molecule while the cluster size increases.

**Low concentrations, at 278,15 K**

Looking at the binding free energies at 278.15 K and low concentration (Figure 5c), the clusters with 1–2 ammonia are again lowest in free energies at larger sizes. This scenario is observed across all cluster compositions. Clusters such as (1,6–7) and (2,7–8) show the lowest binding free energies, indicating that the clusters are more stable with a higher proportion of DMA. Hence, the composition of the lowest free energy clusters are consistent with the 298.15 K and low concentration systems shown in Figure 5a. This could indicate that the concentration and not the temperature is primary driver in determining the lowest free energy cluster compositions. We see a small nucleation barrier in the SA–AM system with a critical cluster of 4 monomers, but no barrier in the other systems.





## High concentrations, at 278,15 K

Figure 5d shows similar trends as the situation at 298.15 K and high concentrations (Figure 5b). Hence, the the lowest free energy composition is the same as at the higher temperature. Obviously, the clusters are lower in free energy compared to the higher temperature. There is seen no nucleation barriers in any of the studied systems with the free energy surface being downhill.

Overall, our results reveal that at low concentrations, the inclusion of DMA in the clusters tends to yield lower, more negative, binding free energies. In contrast, clusters with a higher proportion of AM alone are less stable in the low concentration regime and more stable in high concentration regime. These findings underscore the importance of DMA in the initial *"cluster stabilization"* regime, but also shows the importance of ammonia in facilitating the cluster-to-particle transition, leading to the onset of FNPs in the *"freshly nucleated particle (FNP)"* regime. These results align with the previous hypothesis by Elm et al. (2017); Elm (2017, 2020) that suggested that strong bases like DMA or diamines play a crucial role in the very initial stages of cluster formation, while the subsequent growth is driven by weaker bases such as AM.

## 4 Conclusions

Here we studied the formation of large clusters composed of sulfuric acid (SA), ammonia (AM) and dimethylamine (DMA). Using quantum chemical methods we studied the mixed $(SA)_n(AM)_x(DMA)_{n-x}$ cluster systems, with $n$ from 1 to 10 and $0 \leq x \leq n$ at the $\omega$B97X-D3BJ/6-311++G(3df,3pd)//B97-3c level of theory. We found that the pure SA–DMA clusters are the most stable up to a cluster size of around 8–12 monomers, depending on precursor concentrations, without the need for AM. As the cluster size increases beyond 10–14 monomers, adding 1–3 ammonia molecules significantly increases the stability of the cluster. This suggests a synergistic effect where the presence of a small number of ammonia molecules, in addition to DMA, enhances the overall stability of the sulfuric acid clusters, especially at larger cluster sizes. Additionally, in most of the clusters, AM molecules are embedded in the core, making strong intermolecular interactions with SA, while the DMA molecules reside on the periphery of the cluster. Moreover, we found that the cluster-to-particle transition point in the mixed SA–AM–DMA system occurs at a smaller cluster size of 14 monomers, in contrast to 16 monomers for SA-AM and 20 monomers for SA–DMA found in the previous study by Wu et al. (2024). This suggests a significant synergistic effect when both AM and DMA are present, resulting in the formation of freshly nucleated particles (FNPs) at smaller cluster sizes. The identified base synergy between AM and DMA indicates that nucleation mechanisms are inherently complex and further work is require to study the synergistic effects between other vapours. Hence, additional vapours such as methyl amine (MA) and methane sulfonic acid (MSA) could be interesting to study in the future, as well as the growth of FNPs via uptake of SA, bases and organics.

*Data availability.* All the calculated structures and thermochemistry are available in the Atmospheric Cluster Database (ACDB) (Elm, 2019)





*Author contributions.* Conceptualization: J.E.;

Methodology: G.H., H. W., Y.K., J.E.;

Formal analysis: G.H., H. W., Y.K.;

Investigation: G.H., H. W., Y.K.;

Resources: J.E.;

Writing - original draft: G.H., H. W., Y.K., J.E.;

Writing - review & editing: G.H., H. W., Y.K., J.E.;

Visualization: G.H., H. W., Y.K.;

Project administration: J.E.;

Funding acquisition: J.E;

Supervision: J.E.

*Acknowledgements.* Funded by the European Union (ERC, ExploreFNP, project 101040353). Views and opinions expressed are however those of the authors only and do not necessarily reflect those of the European Union or the European Research Council Executive Agency. Neither the European Union nor the granting authority can be held responsible for them.

This work was funded by the Danish National Research Foundation (DNRF172) through the Center of Excellence for Chemistry of Clouds.

The numerical results presented in this work were obtained at the Centre for Scientific Computing, Aarhus https://phys.au.dk/forskning/faciliteter/cscaa/.





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
