# Peer review of "Base synergy in freshly nucleated particles"

_Aerosol Research, 2024_

## Author Comment (AC1)

**Response to Reviews**

We highly appreciate the positive comments from both the reviewers and all the points have been addressed in the revised paper.

Unfortunately, we spotted a critical error in our script for calculating the free energies at given conditions. This will slightly change Figure 5 and the surrounding text, but have no impact on the overall conclusions of the manuscript (see reviewer # 1 point 8).

We hope that the following responses are satisfying and that the paper can be accepted for publication. The reviewers' comments have been reproduced in blue text below, followed by our point-by-point replies.

**Anonymous Referee #1**

Galib Hasan et al. have studied the formation of freshly nucleated particles from sulfuric acid and two bases, ammonia and dimethylamine, using quantum chemical methods. The manuscript shows the synergetic impact of these two bases on particle formation, and how, despite being a weaker base, larger clusters with addition of one to three ammonia monomers are more stable than clusters consisting purely of sulfuric acid and dimethylamine. The study also bring insight into the transition from clusters to particles, a question that has for long remained open. It is relatively well-written and easy to follow, and the analysis itself is wellmade. Overall,, this study by Galib Hasan et al. furthers our understanding of the formation of freshly nucleated particles, and therefore I recommend it to be accepted for publication after the following comments have been addressed.

**General comments**

The transition of a cluster to a particle is a big part of the main conclusions and relevance of the paper. Therefore, it should be explained more than it is. Now it remains unclear for the reader how exactly the point of transition is defined, and what it is based on.

**Author reply:**

We agree with the reviewer that the cluster-to-particle transition concept should be further elaborated and commented on. We have added the following sentence to the manuscript.

**Added sentence, line 86:**

Specifically, we define FNPs as when one or more ions are fully embedded inside the cluster and when the change in the size-averaged binding free energy approaches zero. Hence, the emergence of FNPs acts as the cluster-to-particle transition point.

In addition, we have added three small sections (2.4, 2.5 and 2.6) to the methods section, further explaining the concept. See the specific comment under point 6. below.

I am also missing some discussion on environmental impacts and the relation of these results to the real atmosphere. For example, conditions with two different temperatures and concentration regimes are studied, and these could be more clearly related to real atmospheric environments. In addition, the differences in the results for these different conditions and their impacts on atmospheric cluster formation could be discussed further.

**Author reply:**

We understand the concern raised by the reviewer and agree that it would be beneficial to better relate the chosen conditions to the real atmosphere. We have added the following text to the manuscript for clarification.

**Added paragraph, line 233:**

To better relate our findings to real atmospheric environments, our studied temperature and concentration regimes can be linked to conditions observed in real atmosphere. For instance, in line with previous studies conducted in Hyytiälä, Finland, which represent typical boreal forest environments, we considered sulfuric acid (SA) concentrations and temperatures that are relevant for new particle formation (NPF) processes. In these environments, SA concentrations are often in the range of  $[SA] = 10^4 - 10^8$  molecules cm-3, with temperatures that generally range between 278.15 K and 298.15 K, depending on seasonal variations. These temperature and concentration conditions closely resemble the boundary layer conditions that are frequently encountered in temperate regions.

There are some typos, formatting issues and minor grammatical issues, and I suggest the authors go over the text with care. Many of these issues are specified in the detailed comments.

**Author reply:**

We have given the paper another thorough read and tried to remove potential formatting and grammar issues.

**Specific comments**

**Introduction**

1. The first paragraph is half a page long, which makes it harder to read. It could be divided into at least two separate paragraphs.

**Author reply:**

We completely agree with the reviewer. The first paragraph has now been broken into three.

2. L37: While this sentence does not really need a reference to begin with, referring to a study over 10 years old does not really fit the point it is trying to make.

**Author reply:**

Indeed, the reference is perhaps not the best fitting one. We have removed the reference.

3. L85: Would it be possible to explain a bit more in detail what this property-based criteria

**for determining the boundary between clusters and particles is?**

**Author reply:**

We agree with the reviewer that the criteria should be further defined. We have added the following sentence to the manuscript.

**Added sentence, line 86:**

Specifically, we define FNPs as when one or more ions are fully embedded inside the cluster and when the change in the size-averaged binding free energy approaches zero. Hence, the emergence of FNPs acts as the cluster-to-particle transition point.

**Methods:**

**4. L134: Please define how this error is determined.**

**Author reply:**

The error was calculated using the DLPNO-CCSD $(T_0)$ /aug-cc-pVTZ binding energy calculations as a reference. We have changed the sentence accordingly:

**Modified sentence, SI:**

**From:**

The error in electronic binding energies is shown in Figure 1.

**To:**

Figure 1 presents the error in the binding energy of several methods, calculated relative to  $DLPNO-CCSD(T_0)/aug-cc-pVTZ$ .

In addition, the entire section has been moved to the SI, as requested by a later comment.

**Results**

5. Figure 3 and the respective text. The only clusters that are labeled are on the upper and lower edges. Therefore, on e.g., L194 (L197), where clusters (0,7) and (0,8) ((0,9) and (0,10)) are discussed, I cannot see what points refer to these clusters and where the free energy values referred to in the text are read from. This should be somehow clarified, the reader should also be able to see where the values come from.

**Author reply:**

We refrained from including too many numbers in the figure to make it less crowded. However, we agree with the reviewer that referring to numbers which are not to be found is perhaps not the best presentation. To circumvent this issue we have instead of using the (0,X) formulation for points containing only SA and DMA in the clusters, explicitly written out these. We have rephrased the following.

**Modified sentence, line 195:**

**From:**

For cluster sizes around 14-16 monomers, compositions such as (0,7) and (0,8) are correspondingly ...

**To:**

For cluster sizes around 14–16 monomers, compositions only containing SA and DMA are correspondingly ...

**Modified sentence, line 198:**

**From:**

This trend is also observed for larger clusters, where the clusters with compositions (0,9) and (0,10) are higher ...

**To:**

This trend is also observed for larger clusters, where the clusters with compositions only containing SA and DMA are higher ...

6. L202: As this transition from cluster to particle is such a big part of the conclusions of the manuscript, the concept and why it is interpreted so should be explained a bit more. What is the threshold value of the averaged binding free energy change from cluster to cluster after which transition to particle is considered to have occurred? And why? In addition, I think a brief explanation on why the leveling out of the averaged binding free energy can be thought of as the cluster transitioning to a particle would also help in clarifying the concept and the relevancy of this paper for the readers who might be less familiar with the topic.

**Author reply:**

We agree with the reviewer that it would be good to give some more background into how exactly the cluster-to-particle transition is defined. To further guide the reader we have added three subsections to the methods that describe the size-averaged binding free energies, the convex hull method and the cluster-to-particle transition in more details.

**Added sections, page 6:**

**2.4 Size-averaged Binding Free Energies**

From the standard binding free energies, we also calculate the size averaged binding free energies  $(\Delta G_{\rm bind}/m)$  of the clusters. The physical interpretation of this quantity can be seen by analyzing  $\Delta G_{\rm bind}/m$  as a function of cluster size. As the cluster size increases a convergence in the  $\Delta G_{\rm bind}/m$  will be seen toward the formation free energy of the bulk system. The physical significance of  $\Delta G_{\rm bind}/m$  can be realized by considering the difference in average binding free energy between a very large  $(SA)_{99}(\text{base})_{99}$  cluster and a  $(SA)_{100}(\text{base})_{100}$  cluster. The addition of one extra acid-base pair would have little impact on the total free energy of the cluster and thereby the gradient of  $\Delta G_{\rm bind}/m$  becomes zero, resembling the bulk particle phase.

**2.5 The convex hull approach**

The emergence of fully coordinated ions in a cluster yield information on the transition from discrete cluster configurations towards the particle phase. In a small cluster all monomers are fully exposed to the exterior and a large stabilization in free energy is gained, when adding more monomers to the cluster. In larger cluster structures fully coordinated ions emerge, corresponding to a "solvated" ion with a solvation shell. Adding more monomers to the existing solvation shell leads to less stabilization free energy gained compared to a smaller cluster.

To investigate when the first fully coordinated ion appears in our calculated cluster structures we here applied the 3-dimensional convex hull approach as described by Wu et al (2024). The applied algorithm is freely available at:

https://gitlab.com/AndreasBuchgraitz/clusteranalysis

**2.6 Cluster-to-particle transition point**

Section 2.4 and 2.5 both yield inferred evidence to the transition from clusters to particles. Using the 3D-convex hull approach we can identify when the first solvation shell is formed in the clusters. Structurally, this implies that we are transitioning from a cluster towards a particle. Thermodynamically, when the size-averaged ( $\Delta G_{\text{bind}}/m$ ) as a function of cluster size becomes constant (the gradient of the change of  $\Delta G_{\text{bind}}/m$  approaches zero), the cluster behaves more like the bulk than a cluster. Hence, we define the cluster-to-particle transition point as when both these conditions are satisfied. I.e. structurally there must be the development of a new phase by having at least one fully coordinated ion and the thermochemistry must resemble the bulk by leveling out in the  $\Delta G_{\text{bind}}/m$  as a function of cluster size. Putting a strict number to when the change in  $\Delta G_{\text{bind}}/m$  resembles the bulk is tricky and most likely system dependent. Hence, tentatively we assign roughly a change of ~1 kcal mol-1 in the  $\Delta G_{\text{bind}}/m$  from cluster size m to m + 1 as the convergence point.

7. L219: "Hence, the cluster-to-particle transition point occurs at a cluster size of 14 monomers..." Please explain how this is determined from Figure 3. In addition, as such a large part of aerosol research is concerned with diameters of clusters/particles, could rough estimations of the diameter these clusters correspond to be given?

**Author reply:**

This is based on that we see the first emergence of a fully coordinated ion and the leveling out in the size-averaged free energies. To clarify, we have added the following sentence.

**Added sentence, line 221:**

These FNP sizes are based on that we see the first emergence of a fully coordinated ion and the leveling out in the size-averaged free energies (see definition in section 2.6).

In addition, as such a large part of aerosol research is concerned with diameters of clusters/particles, could rough estimations of the diameter these clusters correspond to be given?

Indeed, aerosol measurements often refers to the diameter of the clusters/particles. We agree that it would be beneficial to relate the identified cluster-to-particle transition point to the cluster diameters. We have added the following sentence to the manuscript.

**Added sentence, line 222:**

In addition, these cluster-to-particle transition points corresponds to 14.9 Å, 16.7 Å and 17.3 Å for the SA–AM–DMA, SA–AM and SA–DMA systems, respectively.

8. L229: In real world, sulfuric acid concentration also varies between environments. Would including that variation affect the results?

**Author reply:**

Changing the sulfuric acid concentration will most certainly affect the results. To further illustrate this, we calculated Figure 5 at  $SA = 10^7$  molecules cm-3 and  $SA = 10^8$  molecules cm-3. When studying the different concentrations, we unfortunately found a critical error in our script, that implies that our base concentrations were actually much higher than anticipated. To leave minimum impact on the paper we now present the results at  $[SA] = 10^7$  molecules cm-3 results in the main manuscript. We have made minor changes in the surrounding text on page 14-15 to illustrate that we now utilize the  $[SA] = 10^7$  molecules cm-3 data. These changes only affected the critical cluster sizes. Overall, the error will have no major influence on the conclusions drawn. However, we deeply apologize for this crucial mistake.

We added a sentence to the manuscript showing the availability of the data at different SA concentrations in the SI. In addition, we now also describe the effect of SA concentration in each of the scenarios. See page 14-15.

**Added sentence, line 242:**

We also tested the effect of decreasing [SA] to  $10^6$  molecules cm-3 or increasing to  $10^8$  molecules cm-3 (See Figures S3-S5 in the SI).

Technical comments

L46: Formatting issue with reference: "...base molecules Almeida et al. (2013)."

L59: "... in the stabilizing the initial SA clusters"  $\rightarrow$  "... in stabilizing the initial SA clusters"

L155: " ... the GFN1-xTB (Grimme et al., 2017) semi-empirical method."  $\rightarrow$  "... the GFN1-xTB semi-empirical method (Grimme et al., 2017)." When possible, placing the reference to the end of sentence, improves readability of the text.

L180: "We previously Wu et al. (2024) tested ..." should be reformulated e.g., "In Wu et al. (2004), we previously tested ..."

L185: " ... clusters are taken from (Wu et al., 2024),"  $\rightarrow$  "... clusters are taken from Wu et al. (2024),"

L186: Figure 3a, not 2a, I presume.

Figure 3 caption:

" ...  $\leq$  n., under ... "  $\rightarrow$  "...  $\leq$  n, under ... "

"(298,15 k and 1 tm)"  $\rightarrow$  "(298.15 k and 1 atm)"

Figure 4 caption: I would presume yellow is supposed to be sulfur, not nitrogen.

L210: "AM molecule, makes it"  $\rightarrow$  "AM molecule makes it"

L221: "single point refinement single point refinement."

L236: "energies in Figure 5a, shows that"  $\rightarrow$  "energies in Figure 5a show that"

L244: Please define MA and TMA.

L251: "around 4 monomers" Please clarify that this is for the SA-AM.

L235, L245, L257 and L265: "298, 15 K"  $\rightarrow$  "298. 15 K" , "278, 15 K"  $\rightarrow$  "278. 15 K"

**Author reply:**

We highly appreciate the thorough read by the reviewer. All the above technical comments have been corrected accordingly.

**Anonymous Referee #2**

Galib Hasan et al. have conducted an in-depth study on the formation of freshly nucleated particles (FNPs) driven by gaseous sulfuric acid (SA), highlighting the synergistic stabilization effects of significant nitrogen bases, i.e., ammonia (AM) and dimethylamine (DMA) through quantum chemical calculations. Most prior theoretical research has been limited to small clusters, typically those with eight molecules or fewer, focusing on the nucleation stage. The cluster-to-particle transition, a critical yet underexplored process, presents significant challenges for both theoretical and experimental studies. This paper advances our understanding of FNP formation progress, with considerable atmospheric significance. The manuscript is well-organized, data-rich, and analytically rigorous, and hence I am delighted to recommend its publication in Aerosol Research journal after addressing my following comments.

Major Comments:

The study examines the "cluster-to-particle transition point" and concludes that 14 monomers are needed for SA-AM-DMA clusters, 16 monomers for pure SA-AM, and 20 monomers for pure SA-DMA system. However, I am unclear about the criteria used to define the "point" of this transition. It seems not to be solely based on cluster size. So, I kindly suggest the authors clarify this to help other readers avoid similar confusion.

**Author reply:**

We completely agree with the reviewer that this should be better explained. We have significantly expanded the explanation of how we define the cluster to particle transition point. See under reviewer #1 general comments and under point 6.

In the Refined Single Point Energies section, the detailed discussion of errors caused by different calculation methods might be excessive for the main readership of Aerosol Research, who are likely more interested in the environmental impacts. Thus, selectively reducing or relocating some of the detailed content to the Supporting Information may be better, leaving the key results in the main text.

**Author reply:**

Indeed, the section "Refined Single Point Energies" is perhaps a bit too technical for the readership of Aerosol Research. Based on the recommendation, we have moved the entire section to the SI. We have added the following sentence to illustrate that the benchmark is available in the SI.

**Added sentence, line 139:**

We refer to benchmark calculations in the supporting information for a justification of applying the  $\omega$ B97X-DJB3/6-311++G(3df,3pd) level for single point energies.

The authors have systematically studied the SA-AM-DMA system under conditions of low and high temperature as well as low and high concentration, which is highly commendable. Yet, if feasible, relating these findings to specific environmental conditions and discussing their atmospheric significance would further strengthen the environmental implications of the article.

**Author reply:**

We agree that the chosen conditions should be linked better to realitic atmospheric conditions. We have added the following paragraph to the manuscript (see also under reviewer #1, general comments).

**Added paragraph, line 233:**

To better relate our findings to real atmospheric environments, our studied temperature and concentration regimes can be linked to conditions observed in real atmosphere. For instance, in line with previous studies conducted in Hyytiälä, Finland, which represent typical boreal forest environments, we considered sulfuric acid (SA) concentrations and temperatures that are relevant for new particle formation (NPF) processes. In these environments, SA concentrations are often in the range of  $[SA] = 10^{4}-10^{8}$  molecules cm-3, with temperatures that generally range between 278.15 K and 298.15 K, depending on seasonal variations. These temperature and concentration conditions closely resemble the boundary layer conditions that are frequently encountered in temperate regions.

Minor Comments:

Line 52: Dimethylamine  $\rightarrow$  dimethylamine

Line 115: To enhance clarity, please include the full term "Density Functional Theory" when the abbreviation "DFT" is introduced for the first time (line 91).

Please note that in Figure 4's caption, yellow is incorrectly assigned to nitrogen. It should denote sulfur instead.

Line 221: "single point refinement" is repeated twice in the text. Please correct this typographical error.

**Author reply:**

We appreciate the careful read by the reviewer. All the above technical comments have been corrected accordingly.